# The cross-sectional association of parental psychosocial status with children's Body Mass Index z-score and the mediating role of children's energy balance behaviors - the ABCD Study

Meredith L. Overman [1,2,3,4]*, Tanja Vrijkotte[4,5], Yolanda M. Sánchez Castro[3,4], Margreet W. Harskamp-van Ginkel[4,5], Monica Hunsberger[6], Carry M. Renders[4,7], Stef P. J. Kremers[1,2], Mai J. M. Chinapaw [3,4]

1 Department of Health Promotion, Maastricht University, Maastricht, Netherlands, 2 Institute of Nutrition and Translational Research in Metabolism (NUTRIM), Maastricht, Netherlands, 3 Department of Public and Occupational Health, Amsterdam UMC, Vrije Universiteit Amsterdam, Amsterdam, The Netherlands, 4 Amsterdam Public Health research institute, Health Behaviour and Chronic Diseases, Amsterdam, The Netherlands, 5 Department of Public and Occupational Health, Amsterdam UMC, University of Amsterdam, Amsterdam, The Netherlands, 6 School of Public Health and Community Medicine, Sahlgrenska Academy, Institute of Medicine, University of Gothenburg, Gothenburg, Sweden, 7 Department of Health Sciences, Faculty of Science, Vrije Universiteit Amsterdam, Amsterdam, The Netherlands

* m.overman@maastrichtuniversity.nl

**Data Availability Statement:** Due to ethical considerations, the individual data from the ABCD

## Abstract

### Objective

Investigate the cross-sectional association between the psychosocial status of mothers and fathers and the BMI z-scores of their 10 to 12-year-old children. Explore whether this association is mediated by children's diet, physical activity, screen time and sleep. Analyze the moderating effect of the educational levels of both the mother and father on the association.

### Design

In a cross-sectional study design, children's height and weight were measured following a standardized protocol. Parents completed the validated Depression Anxiety and Stress questionnaire, while diet quality, sports participation, time spent in bed and screen time were assessed through child-report using previously validated questions.

### Participants

The data for this study were obtained from the Amsterdam Born Children and their Development study, involving children aged 10 to 12 years and both of their parents (N = 1315).

### Results

The majority, 80%, of the parents were highly educated and born in the Netherlands, and 68% of the children had a healthy BMI. Maternal or paternal psychosocial status was not

study cannot be deposited in a public repository. This decision is guided by the need to protect the privacy and confidentiality of the study participants. The data contains potentially identifying or sensitive information about the participants, and releasing it publicly could breach their privacy rights. The ABCD data are owned by the Amsterdam University Medical Centers, location AMC in Amsterdam, The Netherlands. Any researcher can request the data by submitting a general publication proposal form to the ABCD Project Team as outlined at https://www.amc.nl/web/abcd-studie-2/abcd-studie/voor-onderzoekers-.htm under section use of data (Dutch: Gebruik van data), by email: abcd@amsterdamumc.nl. The ABCD Project Team will check proposals for compatibility with the general objectives, ethical approvals and informed consent forms of the ABCD study, that are in line with the guidelines laid down in the Declaration of Helsinki and all procedures involving research study participants approved by the Central Committee on Research Involving Human Subjects in The Netherlands; the medical ethics review committees of the Academic Medical Center, Amsterdam; the VU University Medical Center Amsterdam; and the Registration Committee of the Municipality of Amsterdam. There are no other restrictions to obtaining the data and all data requests will be processed in the same manner.

**Funding:** This research is part of the Lifestyle Innovations Based on Youth Knowledge and Experience (LIKE) program and is supported by a grant from the Netherlands Cardiovascular Research Initiative: an initiative with support of the Dutch Heart Foundation, ZonMw, CVON2016-07 LIKE, and Sarphati Amsterdam. The funders had no role in study design, data collection and analysis, decision to publish, or preparation of the manuscript.

**Competing interests:** The authors have declared that no competing interests exist.

significantly associated with children's BMI z-score (maternal β -0.0037; 95% CI: -0.008 to 0.0007, paternal β 0.0028; 95% CI: -0.007 to 0.002). Screen time mediated the association between paternal psychosocial status and children's BMI z-score (β = 0.010, 95% CI: 0.002; 0.020). Children's diet, physical activity, and sleep did not mediate the association between paternal psychosocial status and children's BMI z-score. Parental educational level was not a moderator.

## Conclusions

This research is unique in including four energy balance behaviors and including both mothers and fathers' psychosocial status. Children withfathers experiencing poorer psychosocial status engaged in more screen time which partly explained their higher BMI z-score.

## 1. Introduction

Childhood overweight and obesity are global public health issues. Behavioral determinants of overweight and obesity are the consumption of energy dense food and beverages, lack of physical activity, lack of sleep and excessive screen time [1–5]. These behaviors are, in turn, influenced by the interplay between environmental and individual factors [6]. Parents are part of their children's social environment and influence above mentioned energy balance behaviors, for example, through parenting practices and creating a more or less healthy home environment [7].

Previous research found that parental psychosocial status, defined as the interplay between depression, anxiety and stress levels, was associated with childhood energy balance behaviors [7–11] or BMI [7,12,13]. Children, whose parents had a higher stress level, had less opportunities for physical activity than children with less stressed parents [14]. Furthermore, higher levels of parental stress has been associated with lower TV restrictions for children [14,15]. Maternal, depressive symptoms have been associated with children's sleep/wake problems and paternal depressive symptoms with shorter time in bed of children and fewer sleep minutes [16]. Zarychta, et al. [9] found that higher levels of parental depression were associated with higher levels of child BMI, which was mediated by physical activity. Shankardass et al. [11], found that higher physical activity and higher parental education partially explained the relationship between parental stress and preadolescent BMI.

No previous studies have examined the interplay between depression, anxiety and stress, of both mother and father separately in relation to children's energy balance behaviors and BMI z-score. Although both mother and father play a pivotal role in guiding their children to a healthy diet, physical activity, and sleep [17], childhood obesity research has predominantly focused on the mother's role [18–20]. Therefore, the current study explores the independent association of both maternal and paternal psychosocial status with children's BMI z-score. Furthermore, we examined whether this association was mediated by children's dietary intake, sports participation, time in bed and screen time. As parents with higher levels of education tend to have children with healthier energy balanced-behaviors [21,22] and lower BMI, we also examined the potential moderating (i.e. effect modifying) role of parental educational level.

We hypothesized that increased parental depression, anxiety and stress, whether experienced by the mother or father, would be associated with increased BMI z-scores in their

children. Further, we hypothesized that this association is partly mediated by children's dietary intake, sports participation, time in bed and screen time. Finally, we hypothesized that the association between parental psychosocial status and children's energy balance-related behavior is stronger among children from parents with a lower educational level.

## 2. Methods

### 2.1 Study design

We utilized cross-sectional data from phase 4 of the Amsterdam Born Children and their Development (ABCD) study collected in 2015. The ABCD study is a prospective ongoing population-based cohort study that examines the association between maternal lifestyle, health, psychosocial and environmental circumstances during pregnancy, and children's health at birth and later in life (http://www.abcd-study.nl/). Details about the ABCD study, including its design, conceptual framework and measurements have been described previously [23]. Phase 4 of the ABCD study focused on children 10 to 12 years old and both their parents.

This study was conducted according to the guidelines laid down in the Declaration of Helsinki and all procedures involving research study participants were approved by the Central Committee on Research Involving Human Subjects in The Netherlands; the medical ethics review committees of the Academic Medical Center, Amsterdam; the VU University Medical Center Amsterdam; and the Registration Committee of the Municipality of Amsterdam. The Ethics approval numbers for the different parts of the ABCD-study are: (1) Pregnancy questionnaire: METC AMC 02/039#02.17.392; (2) Questionnaires and health check at age 5–6: METC AMC 02/039#07.17.1039; (3) Questionnaires and health check at age 11–12: METC AMC 2015_154#B2015655a.

Written informed consent was obtained from each mother upon enrollment in the study and again at age 5, as part of preventive health monitoring by the ministry of health. Additionally, written informed consent was obtained from the child starting at age 12 and participating fathers.

The study has been reported in accordance with the Strengthening the Reporting of Observational Studies in Epidemiology–Nutritional Epidemiology (STROBE-nut) guidelines for nutritional epidemiological research guidelines [24].

### 2.2 Study population

In 2002 and 2003, all pregnant women in Amsterdam were invited to participate in the ABCD study during their first appointment with the obstetrician. Subsequently, mothers were asked for permission to track their child's development by sending invitations every five years. In relation to phase 4, the first questionnaire was filled out on March 5th, 2015, and the father questionnaire (the most recent one) was filled out by May 21st, 2017. Physical measurements were taken during this follow-up period.

Phase 4 marked the first-time fathers were invited to participate. In total, 3018 children, 2997 mothers, and 2264 fathers completed a questionnaire (Fig 1). Data on BMI were obtained from ABCD's physical examination [23] or if unavailable the Amsterdam Youth Health Care data. Mother-father-child trios with complete data for both parents and child were included. We included all parents who filled in the mother and father's questionnaire, regardless of whether they were the biological parent or not. Our study does not include children with same sex parents or from single households. The final sample size included 1315 mother-father-child trios.

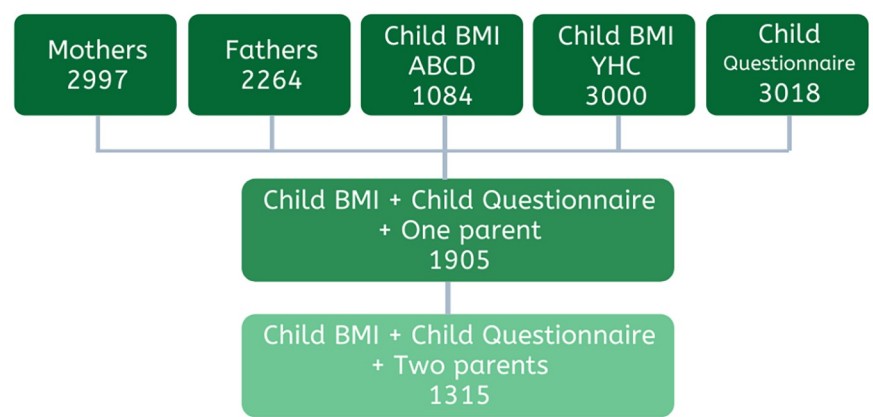

**Fig 1. ABCD phase 4 data selection.**

### 2.3 Measurements

**2.3.1 Children's Body Mass Index (BMI) z-score.** Children's height and weight were measured according to a standardized protocol by trained nurses at the Youth Health Care center (n = 1503), or the ABCD health check (n = 590). In case both measurements were available, the ABCD measurement was used since it was closest to the questionnaire data. Each Child's Body Mass Index was calculated (BMI z-score; calculated as weight (kg)/height (m2)) and converted into BMI z-scores, using age and sex adjusted reference tables [25].

**2.3.2 Parental psychosocial status.** Both mother and father completed the validated Depression Anxiety and Stress questionnaire (DASS-21) with 21 items regarding their feelings and behaviors related to psychological issues [26]. Depression was assessed with questions such as "*I did not experience any positive feeling*", Anxiety, with questions such as, "*I experience breathing difficulty (e.g., excessively rapid breathing, breathlessness in the absence of physical exertion)*" and for stress, questions such as "I found it difficult to relax" were included. The questions were answered on a four-level scale, from (almost) never (0) to very often (3). We computed a total score of the DASS 21 including all three psychopathologies per parent. Since depression, anxiety and stress are correlated moderately high, using the total parental psychosocial status score gives a general impression of parental psychological status [26,27]. The total score ranges from 0 to 63, from no to severe psychological distress.

### 2.4 Potential mediators

All energy balance-related behaviors were assessed by child report.

**2.4.1 Diet quality score.** The ABCD study selected relevant items from existing questionnaires to assess children's food intake [28,29]. The dietary intake questionnaire covers four food groups: fruits (excluding juice), vegetables, snacks (e.g. biscuit, candy, chocolate, crisps) and sugar-sweetened beverages (SSBs) (i.e. fruit juice, soft drinks). Fruit and vegetable intake were assessed in servings per day. Snack intake was assessed in portions per day. Children were asked how many glasses of SSBs they drank on average during a school day and a

weekend day. For each food group, the number of servings per day was divided into quartiles. Each quartile was rated 1 to 4. Based on that, a total Diet Quality Score was calculated. The minimum score was four representing an unfavorable consumption of SSB, snacks, fruits and vegetables. The maximum score was sixteen representing a favorable consumption of SSB, snacks, fruits and vegetables. Missing scores were imputed if at least two dietary components were available. The calculation of the Diet Quality Score is in line with a study conducted as part of the broader ABCD study [30].

**2.4.2 Sports participation.** Sports participation included the items "*do you practice any sport*?", *"If yes, how many hours do you practice this sport per week? (including training and matches)"*. If the child played more than one organized sport the same question was repeated up to three times. Based on these items, a MET-score was calculated by multiplying the total hours of sports participation with the metabolic equivalent for that sport [31]. If the child reported more than one sport, the same procedure was followed and then added up to obtain a final score.

**2.4.3 Time in bed.** For time in bed children answered the following questions: "*At what time do you usually go to bed*?" and *"At what time do you usually get up?"* separately for school days and weekends. The average time in bed on a school day was multiplied by 5 and the average time in bed on the weekends was multiplied by two. Subsequently, we calculated the average number of hours spent in bed.

**2.4.4 Screen time.** Screen time included two questions: "How long do you usually watch TV in your spare time, per day?" and "How long do you usually use a computer, game console, tablet, mobile, and laptop, per day?". These questions were answered separately for school days and weekends. The average screen time on a school day was multiplied by 5 and the average screen time on the weekends was multiplied by two. Following that, we computed the average number of hours of screen time.

## 2.5 Moderator—maternal and paternal education

Mothers and fathers self-reported their highest level of education, which we categorized as low (primary school/technical secondary education or lower vocational secondary school), medium (degree in higher vocational secondary education/academic secondary education/ intermediate vocational education) or high (degree in higher vocational education/university).

## 2.6 Covariates

In the respective analyses, the place of birth of the mother or father (Netherlands, Other Western, Non-western country) served as a covariate. Parental BMI was calculated as weight (kg)/ height (m2) from self-reported data.

## 2.7 Statistical analyses

The final analysis includes complete mother-father-child trios and was performed using IBM SPSS package version 26.0 (SPSS Inc., Armonk, NY, USA). Covariates were organized in a DAG using DAGGity and a minimal adjustment set was identified: parental BMI; and DASS score of the other parent; (see Figs 2 and 3) [32]. First, we examined the association of maternal and paternal psychosocial status, respectively, with child's BMI z-score using linear regression (total effect). Second, we performed a multivariable mediation analysis including the potential mediators: diet quality score, sports participation, time in bed and screen time using model 4 of the PROCESS macro with 5000 bootstrap samples for calculating 95% confidence intervals [33]. Multiple mediation models separate the direct and indirect effect of X and Y through the different mediators, with the condition of no causal influence between mediators. Both models

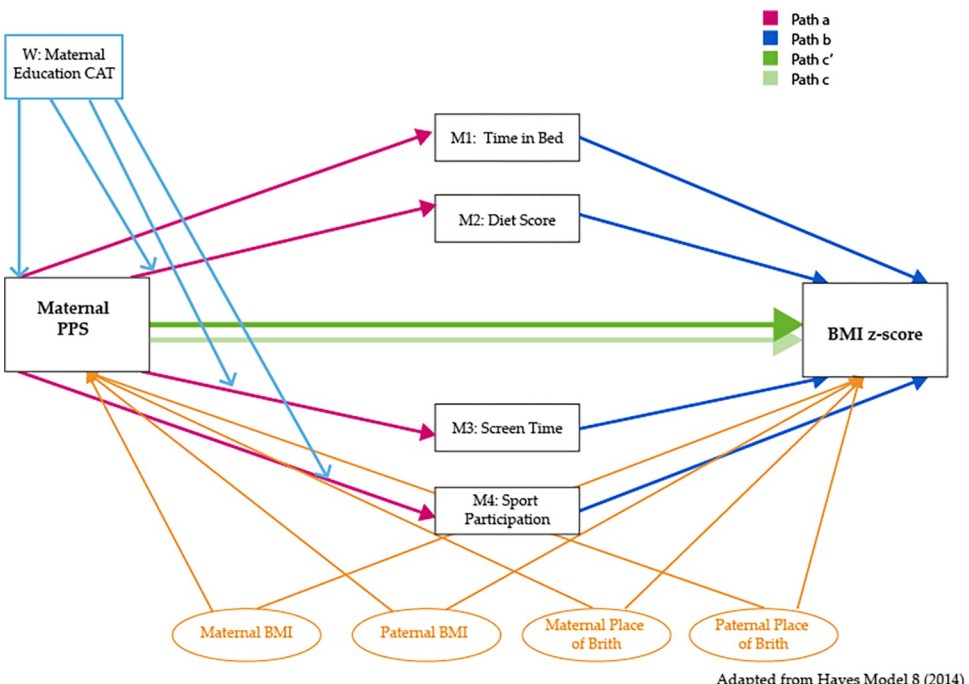

**Fig 2. Association between maternal psychosocial status and child BMI including moderation by maternal educational level and mediation by energy balance behaviors.**

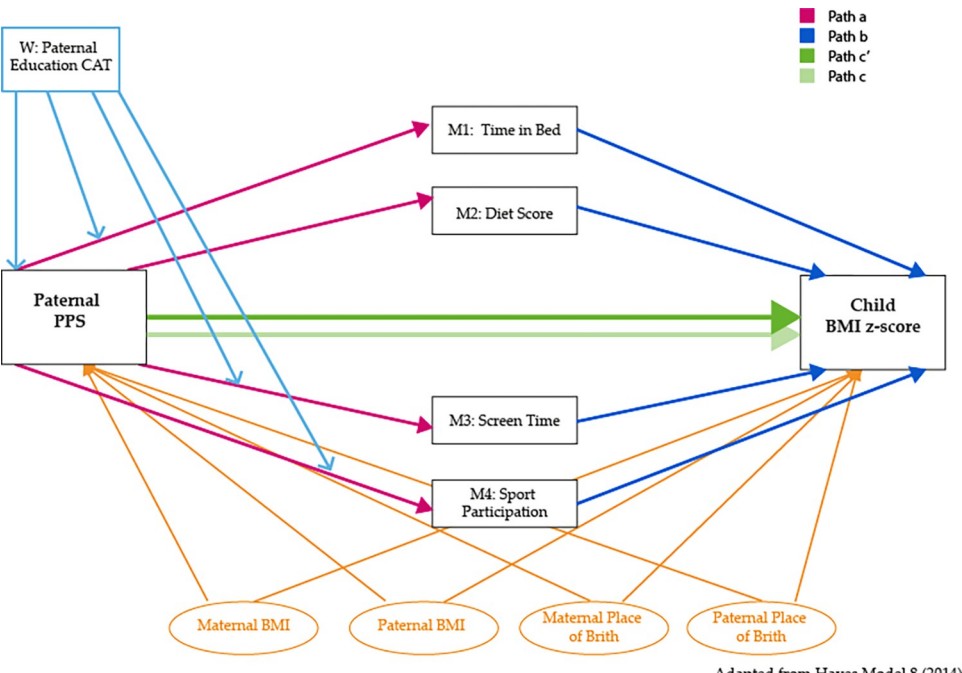

**Fig 3. Association between paternal psychosocial status and child BMI including moderation by paternal educational level and mediation by energy balance behaviors.**

were adjusted for place of birth of the parent, parent BMI and the counter parent's psychosocial status. In the final step, we conducted tests for moderated mediation, with the educational level of the respective parent serving as the moderator (i.e. effect modifier) in the association between parental psychosocial status and the potential mediators. In case of a significant interaction between parental psychosocial status and educational level (p<0.10), we performed a stratified mediation analysis using model 8 of Andrew Hayes [34]. Hayes moderated mediation analysis works through *conditional processing analysis*. Hayes' methods combined the estimation of the direct and indirect effects of parental educational level on children's BMI z-score, through the proposed mediators followed by an inferential test for such indirect effects. We did not adjust for multiple testing but instead report effect sizes, confidence intervals, and p-values of all performed analyses to enable readers using their own judgment about the relevance of the observed associations. Furthermore, in this study, we consider mediation even if the association between the main exposure and outcomes is not statistically significant, as mediation analyses provide a more complete picture of associations by explaining working mechanisms [35].

## 3. Results

Table 1 presents the characteristics of the study sample. In total, 1315 mother-father-child trios had complete data on children's BMI z-score, dietary intake, sports participation, time in bed and screen time. The mean BMI z-score of the children was -0.27 and 10% of the children had overweight. The mean diet quality score was 8.3 ranging from 4 to 16. With regard to sports participation, the mean value was 28.6 MET/hours per week. The mean hours in bed were 10.5, and mean screen time 3.1 hours per day, ranging from 1 to 10 hours per day. Twenty-three percent of the mothers and 45% of the fathers had overweight. Furthermore, 81% of the mothers and 75% of the fathers had a high educational level. The mean psychosocial status score was 11.3 for mothers and 10.7 for fathers.

### 3.1 Multiple mediation analysis

Tables 2 and 3 present the association of maternal and paternal psychosocial status, respectively with children's BMI z-score, and potential mediation by children's diet, sports participation, time in bed, and screen time. Neither maternal psychosocial status (β -0.0037; 95% CI: -0.008 to 0.0007) nor paternal psychosocial status (paternal β 0.0028; 95% CI: -0.007 to 0.002) were significantly associated with children's BMI z-score. Maternal psychosocial status was not significantly associated with any of the four energy balance behaviors. Poorer paternal psychosocial status was associated with more screen time (β 0.009; 95% CI: 0.003 to 0.016), but not with the other energy balance behaviors. Three out of four mediators were significantly associated with BMI-z-score, although associations were small: less time in bed (β -0.126; 95% CI: -0.216 to -0.036), a higher diet quality score (β 0.046; 95% CI: 0.023 to 0.688) and more screen time (β 0.090; 95% CI: 0.056 to 0.128) were associated with a higher BMI-z-score. Screen time was a significant mediator of the association between paternal psychosocial status and children's BMI z-score (a*b: β = 0.010 (95% CI: 0.002; 0.020). No other significant mediation effects were found. Parental educational level did not moderate the association between parental psychosocial status and children's energy balance-related behaviors.

## 4. Discussion

This cross-sectional study provides insight into the association between maternal psychosocial status and children's BMI z-score, independently from paternal psychosocial status and vice versa. Parental psychosocial status was not significantly associated with children's energy

**Table 1. Characteristics of children, their mothers and fathers: Amsterdam born children and their development cohort study.**

| | | N | Mean (SD) |
|---|---|---|---|
| Children | | | |
| Age, years | | 1315 | 11.4 (0.7) |
| BMI z-score | | 1315 | -0.27 (1.0) |
| BMI categorical | Underweight | 289 | 22% |
| | Normal weight | 891 | 67.8% |
| | Overweight | 135 | 10.3% |
| Time in Bed | (Hrs/24 hours) | 1315 | 10.5 (0.5) |
| Diet Quality Score | (4–16 Pts) | 1315 | 8.3 (2.2) |
| Screen Time | (Hrs/day) | 1315 | 3.1 (1.4) |
| Sport Participation | (MET hours/week) | 1315 | 28.6 (14.9) |
| Sex | Girls | 659 | 50.1% |
| Mother | | | |
| Place of birth | Netherlands | 1133 | 86.2% |
| | Other Western Country | 84 | 6.4% |
| | Other Non-Western Country | 98 | 7.5% |
| Mother BMI[a] | Underweight | 31 | 2.4% |
| | Normal weight | 977 | 74.3% |
| | Overweight | 307 | 23.3% |
| Psychosocial Status | (0–63 Pts) | 1315 | 11.3 (11.2) |
| Educational Level | Low and Medium | 246 | 18.7% |
| | High | 1069 | 81.3% |
| Father | | | |
| Place of birth | Netherlands | 1135 | 86.3% |
| | Other Western Country | 66 | 5.0% |
| | Other Non-Western Country | 114 | 8.7% |
| Father BMI[a] | Underweight | 5 | 0.4% |
| | Normal weight | 719 | 54.7% |
| | Overweight | 591 | 44.9% |
| Psychosocial Status | (0–63 Pts) | 1315 | 10.7 (11.6) |
| Educational Level | Low and Medium | 320 | 24.3% |
| | High | 995 | 75.7% |

[a]BMI = Body Mass Index.

balance behaviors nor their BMI z-score. However, time in bed, dietary quality and screen time were significantly associated with children's BMI z-score. Only screen time appeared a significant mediator in the association between paternal psychosocial status and children's BMI z-score. Children of fathers with 10 points higher psychosocial score had 5 min/day more screen time, and children with 1 hour/day more screen time had 0.09 points higher BMI z-score. Parental educational level did not moderate the association between parental psychosocial status and children's energy balance-related behaviors.

Most parents in the current study were highly educated (81% of the mothers and 76% of the fathers). Since higher educated parents tend to have fewer psychosocial complaints this may explain the lack of significant associations,. Moreover, children from highly educated parents tend to engage in more physical activity, less screen time, have a higher intake of fruit, vegetables [21,22] and a lower BMI [36]. Previous studies found that children from parents with low

**Table 2. The association between maternal psychosocial status and child BMI z-score and potential multiple mediation by energy balance behaviors (n = 1315).**

| | Maternal PS[a] → Mediator | Mediator → BMI[b] z-score | Mediation → |
|---|---|---|---|
| Mediator | a-path (bootstrapped 95% CI) | b-path (bootstrapped 95% CI) | (a x b) (bootstrapped 95% CI) |
| Time in bed | β = -0.0001 (95% CI: -0.0028; 0.0026) | **β = -0.126 (95% CI: -0.216; -0.036)** | β = 0.0001 (95% CI: -0.0038; 0.0045) |
| Diet quality score | β = 0.004 (95% CI: -0.007; 0.014) | **β = 0.046 (95% CI: 0.023; 0.688)** | β = 0.002 (95% CI: -0.004; 0.008) |
| Screen Time | β = -0.002 (95% CI: -0.009; 0.005) | **β = 0.090 (95% CI: 0.052; 0.128)** | β = -0.002 (95% CI: -0.010; 0.005) |
| Sport participation | β = -0.053 (95% CI: -0.126; 0.020) | β = 0.002 (95% CI: -0.001; 0.006) | β = -0.001 (95% CI: -0.005; 0.001) |
| indirect effect | | β = -0.0016 (95% CI: -0.012; 0.009) | |
| direct effect (maternal PS → BMI z-score) | | β = -0.0036 (95% CI:-0.008; 0.001) | |
| Total Effect | | β = -0.0037 (95% CI:-0.008; 0.0007) | |

[a]PS = psychosocial status

[b]BMI = Body Mass Index.

Full model adjusted for Paternal BMI, Maternal BMI, Paternal psychosocial status, place of birth of the parent.

Bold indicates a significant association.

levels of anxiety and stress were also more likely to engage in healthy habits. In our sample we indeed found a significant association between parental psychosocial status and screen time.

The sample characteristics reflect that our sample was relatively healthy. For example, most parents had normal levels of psychosocial status according to the DASS-21 score [26]. Similarly, the four energy balance behaviors were on average in line with WHO recommendations [1].

Regarding the association between parental psychosocial status and children's BMI, our findings are in line with Walton et al. [14] who found no significant association between parental stress and children's BMI z-score either. Other studies found a significant association of parental depression or stress, with children's BMI [7,9,11–13]. Regarding the association between parental psychosocial status, and energy balance-related behaviors, results of previous studies are inconsistent. Parks et al. found no significant association between parent's stressors or parent-perceived stress and children's physical activity. Consistent with our findings, neither of these studies found an association between parental stress and dietary intake of the children [10,37]. El Sheikh [16] found that higher parental depressive symptoms was associated with sleep/awake problems and shorter time in bed, which was contrary to our findings. Some studies found that higher stress levels of parents were associated with lower physical activity,

**Table 3. The association paternal psychosocial status and child BMI z-score and potential multiple mediation by energy balance behaviors (n = 1315).**

| | Paternal PS[a] → Mediator | Mediator → BMI[b] z-score | Mediation |
|---|---|---|---|
| Mediator | a-path (bootstrapped 95% CI) | b-path (bootstrapped 95% CI) | (a x b) (bootstrapped 95% CI) |
| Time in bed | β = -0.0001 (95% CI:-0.0027; 0.0025) | **β = -0.126 (95% CI: -0.216; -0.036)** | β = 0.0002 (95% CI: -0.0046; 0.0049) |
| Diet quality score | β = -0.009 (95% CI:-0.019; 0.002) | **β = 0.046 (95% CI: 0.023; 0.688)** | β = -0.005 (95% CI: -0.011; 0.001) |
| Screen Time | **β = 0.009 (95% CI: 0.003; 0.016)** | **β = 0.090 (95% CI: 0.052; 0.128)** | **β = 0.010 (95% CI: 0.002; 0.020)** |
| Sport participation | β = -0.046 (95% CI: -0.116; 0.025) | β = 0.002 (95% CI: - 0.001; 0.006) | β = -0.001 (95% CI: -0.005; 0.001) |
| indirect effect | | β = 0.0038 (95% CI: -0.008; 0.016) | |
| direct effect (paternal PS → BMI z-score) | | β = -0.0031 (95% CI: -0.007; 0.001) | |
| Total Effect | | β = 0.0028 (95% CI: -0.007; 0.002) | |

[a]PS = psychosocial status

[b]BMI = Body Mass Index.

Full model covariates: Paternal BMI, Maternal BMI, Maternal psychosocial status, place of birth of the parent.

Bold indicates a significant association.

less screen time restrictions and higher sedentary behavior [10,14,15]. Less rules and restrictions can result in more screen-based or sedentary behaviors among children [15,38]. In our study, screen time was indeed a mediator in the association between paternal psychosocial status and BMI z-score.

In line with previous literature, less time in bed and more screen time were associated with higher child BMI z-scores [1,2,4,5]. Contrary to our hypothesis and previous literature, we found that children with a better diet quality score had a higher BMI z-score. This might be explained by children's social desirable answering [39]. Peer influence, body image concerns and diet intake become more relevant for pre-adolescences, resulting in under and over reporting of dietary behaviors [40]. Another explanation may be that children with a higher BMI z-score are more conscious of their dietary behaviors or parents may set more rules to prevent further increases in the BMI z-score. Sport participation was not significantly associated with children's BMI z-score. One explanation may be that we did not include all types of physical activity contributing to energy expenditure such as active transport or outdoor play.

## 4.1 Strengths and limitations

A major strength and distinctive aspect of our study is that we have data on the psychosocial status of both mothers and fathers. In research fathers are often underrepresented. Another major strength is the substantial sample size. 1,315 mother-father-child trios is more than in other comparable studies on the association between the psychosocial status of mothers or fathers and the BMI of their children [7,9,10,12]. Moreover, height and weight of the children were measured by trained personnel according to standardized procedures. BMI remains a valuable screening tool, especially at the population level, to identify individuals potentially at risk of being overweight or underweight. We used the BMI z-score i.e. BMI adjusted for age and sex of the child. However, it is important to acknowledge that BMI also has limitations. BMI lacks precision in distinguishing between muscle and fat mass and our reference values were not specific for ethnicity. Although sports participation is not a perfect measure of overall physical activity, organized sports participation is associated to overall moderate to vigorous physical activity as well as compliance with physical activity guidelines [41,42]. In addition, we did not consider parents' dietary habits and physical activity. Hence, our understanding of shared family dynamics related to energy balance behaviors is limited. Furthermore, we used frequently used validated questionnaires, enabling comparison with previous studies. However, this study utilized the Diet Quality Score to assess the consumption of fruits, vegetables, snacks, and sugar-sweetened beverages which has not been validated to date. However, the Diet Quality Score includes pivotal elements of children's dietary pattern that have been shown to be associated with future health [28,29]. Moreover, this self-report of energy balance behaviors may suffer from recall bias and social desirable answering, which might have led to over- and underreporting. This can be considered a limitation. Furthermore, the cross-sectional design of the ABCD study prohibits conclusions regarding a temporal causal relationship between variables. Another limitation is that we did not include data from households with single parents or with same-sex parents. Moreover, we lacked information on biological parenthood. Nonetheless, despite these limitations, this research is unique in including four energy balance behaviors and including both mothers and fathers psychosocial status.

## 4.2 Recommendations

We observed a significant association between paternal psychosocial status and screen time, and between screen time and children's BMI z-score. However, more research is needed to confirm and further explain the association between paternal psychosocial status and

children's BMI z-score through screen time. Future research should be carried out in more diverse, at risk groups in order to understand how parental psychosocial status might affect children's BMI z-score through energy balance-related behaviors.

## 5. Conclusion

In conclusion, children with fathers experiencing poorer psychosocial status engaged in more screen time which partly explained a higher BMI z-score. We found no other mediating effects of energy balance behaviors, and there were no other significant associations between parental psychosocial status and children's BMI z-score.

## Supporting information

**S1 Checklist.**
(DOCX)

**S1 Table. STROBE-nut: An extension of the STROBE statement for nutritional epidemiology.**
(DOCX)

## Acknowledgments

The authors thank all participating mothers, fathers and their children for their time and involvement in the ABCD cohort study. In addition, we thank Roel Hermans, PhD, who provided valuable feedback on the manuscript in an early stage.

## Author Contributions

**Conceptualization:** Meredith L. Overman, Tanja Vrijkotte, Yolanda M. Sánchez Castro, Monica Hunsberger, Mai J. M. Chinapaw.

**Formal analysis:** Yolanda M. Sánchez Castro.

**Writing – original draft:** Meredith L. Overman, Yolanda M. Sánchez Castro, Mai J. M. Chinapaw.

**Writing – review & editing:** Meredith L. Overman, Tanja Vrijkotte, Yolanda M. Sánchez Castro, Margreet W. Harskamp-van Ginkel, Monica Hunsberger, Carry M. Renders, Stef P. J. Kremers, Mai J. M. Chinapaw.

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
