## [Decision Letter · Decision Letter 0]

10 Dec 2023

PONE-D-23-29554The association of parental psychosocial status with children’s Body Mass Index and the mediating role of children’s energy balance behaviors - the ABCD StudyPLOS ONE

Dear Dr. Overman,

Thank you for submitting your manuscript to PLOS ONE. After careful consideration, we feel that it has merit but does not fully meet PLOS ONE’s publication criteria as it currently stands. Therefore, we invite you to submit a revised version of the manuscript that addresses the points raised during the review process.

We look forward to receiving your revised manuscript.

Kind regards,

Henri Tilga, PhD

Academic Editor

PLOS ONE

Additional Editor Comments:

The Reviewers have provided several useful comments to increase the quality of this manuscript. Please carefully follow all the comments made by the Reviewers and revise the manuscript accordingly.

Reviewers' comments:

Reviewer's Responses to Questions

**Comments to the Author**

1. Is the manuscript technically sound, and do the data support the conclusions?

Reviewer #1: No

Reviewer #2: No

2. Has the statistical analysis been performed appropriately and rigorously? 

Reviewer #1: No

Reviewer #2: No

3. Have the authors made all data underlying the findings in their manuscript fully available?

Reviewer #1: Yes

Reviewer #2: No

4. Is the manuscript presented in an intelligible fashion and written in standard English?

Reviewer #1: Yes

Reviewer #2: No

5. Review Comments to the Author

Reviewer #1: Dear PLOS ONE editor and authors

I've commented on the manuscript for improving readability.

General comments

The main issue with this manuscript is that statistical analyses do not match the study's aim, and the results do not support the authors' conclusion. I've described the reasons below.

Title

1. I prefer using "The cross-sectional association of ..." instead of " The association of ...".

Abstract

2. Please indicate the results of mediation analyses on children’s diet, physical activity, and sleep.

Introduction

3. I disagree that educational level serves as a potential mediator. Parent educational levels are just a confounder, not a mediator. Current psychological stress never causes low socio-economic status.

4. In addition to the above comment, the authors appear to confuse mediation with effect modification. Please revise sentences in the introduction section distinguishing these concepts.

Materials and methods

5. If the dietary quality score has been validated, please refer to a corresponding paper. If not, please describe the limitation of nonvalidating in the discussion section.

6. Please explain why children's age and sex were not included as covariates.

7. In addition to comment 6, there can be other potential confounders. Referring to previous studies, please confirm covariates in this study are sufficient for analyses.

8. Paternal and maternal psychosocial status should be adjusted simultaneously. This is because mothers whose partners have poorer psychosocial status tend to exhibit worse psychosocial status, the effect of paternal psychosocial status can include maternal ones, and vice versa.

9. The authors have a typo, misspelling "SPSS" as "SPS."

Results

10. Because the main exposures were not significantly associated with the outcome, the interpretation of the subsequent mediator analyses is questionable. If such interpretation is permissible, the authors must cite supporting literature.

Discussion

11. The argument of this paper seems inconsistent with the results. Considering the lack of a significant association and the small beta from linear regression models, it would be accurate to conclude that there was no association between parental psychosocial status and children's Body Mass Index.

Reviewer #2: The authors investigated if there is an association between parental depression, anxiety, or stress symptoms are associated with their children’s (aged 10 - 12 years) body mass index (adjusted for age and sex). The study was performed using data from 1,315 children and their parents participating in the Amsterdam Born Children and their Development (ABCD) study.

Unfortunately, I cannot recommend this manuscript for publication as I firmly believe there are fundamental issues with the study measures used to assess diet quality and physical activity and the statistical analyses performed. Furthermore, the small sample size and lack of replication mean that the publication of these results would only contribute to the reproducibility problem and further erode the public’s trust in scientific research, without providing any positive impacts to the ABCD Study participants, the general public, or the medical and scientific communities.

Below I have provided a few general comments before listing my concerns with the study.

General comments:

• BMI is a different measurement to weight and using these terms interchangeably (for example in the short title) is both factually incorrect and misleading.

• The use of very long sentences throughout makes this manuscript hard to read.

• There are grammatical, nomenclature, and English language mistakes throughout the manuscript.

• It was not mentioned if adjustment for multiple testing was performed.

Concerns:

• The BMI mathematical formula was devised based on the average Western European man’s physical characteristics and does not (1) distinguish between excess body fat, bone mass, or musculature, (2) interpret the distribution of fat (which is a predictor of negative health outcomes such as type 2 diabetes and heart disease), and (3) cannot distinguish between sex, age, or ethnicity.

o The authors highlight that this is a muti-ethnic study, however, the BMI formula’s inability to distinguish between ethnicities has not been addressed in the study. Also age and sex adjusted reference tables were used to create the z-scores for the children, however, this was not done for the parents.

o Furthermore, there is an ever-growing body of evidence showing that BMI should not be used as a standalone tool for categorising individuals as overweight or obese (or getting a full overview of an individual’s overall health). Rather, multiple measures should be considered, including: body adiposity index, waist circumference, and waist-to-hip index.

• The dietary quality score used in the study does not include the five standard food groups. Therefore, a comprehensive overview of the child’s diet is not used by the study.

• The study only included sports participation as a measure of physical activity. This oversimplifies the possible physical activities the child could be performing on a regular basis, such as riding their bicycle or running. Furthermore, the parents’ diet or activity levels were not included, which is a massive limitation as shared family dynamic was not investigated.

• Although the study has data on full mother-father-child trios the analyses were performed on parent-child dyads. The authors state that: “We included all parents who filled in the mother and father’s questionnaire, regardless of whether they were the biological parent or not. Our study does not include children with same sex parents or from single households.”. I have concerns about the authors choice to conduct analyses separately for each parent, rather than looking at the complete family. Furthermore, sensitivity analyses should have been performed for non-biological relationships to disentangle the direct and indirect familial effects. Finally, the exclusion of children from households with single parents and same-sex couples prevents the results to be generalisable to the general population.

• The authors state “Another major strength is the considerable sample size.”, however, the study included only 1,315 mother-father-child trios. Given this small sample size, I have considerable concerns about the statistical power the study has to identify a true effect of parental psychosocial status with children’s BMI (or the potential study mediators: children’s diet, sports participation, time in bed, and screen time). Furthermore, no replication analyses were performed, therefore, no evidence that the findings are reliable and applicable to the population as a whole were provided.

Reviewer Confidential Comments to Editor

Unfortunately, I do not recommend this manuscript for publication as I believe there are fundamental issues with the study measures and statistical analyses. Furthermore, the sample size and lack of replication mean that the publication of these results would only contribute to the reproducibility problem and further erode the public’s trust in scientific research, without providing any positive impact to the ABCD Study participants, the general public, or the medical and scientific communities.

6. PLOS authors have the option to publish the peer review history of their article (what does this mean?). If published, this will include your full peer review and any attached files.

Reviewer #1: No

Reviewer #2: No

---

## [Author Response · Author response to Decision Letter 0]

11 Mar 2024

Subject: Re: PONE-D-23-29554 - Submission of Revised Manuscript

Dear Dr. Tilga and reviewers, 

We appreciate the opportunity to revise and resubmit our manuscript. Thank you for taking the time to review our manuscript and for your valuable questions and feedback regarding the manuscript. We appreciate your thorough review and we have carefully reviewed the comments the academic editor and reviewers provided and addressed each point thoroughly in our revised submission.

I have uploaded the revised submission to the submission system and included the following items:

1. Response to reviewers: A detailed rebuttal letter addressing each point raised during the review process.

2. Revised Manuscript with Track Changes: A marked-up copy of the manuscript highlighting the changes made from the original version.

3. Manuscript: An unmarked version of the revised paper without tracked changes.

4. Strobenut: with revised page numbers and line numbers. 

5. Revised Cover letter. 

6. Figures, which were checked by PACE software. 

We have also ensured that our financial disclosure statement and data availability is updated in the cover letter. 

Thank you once again for your time and consideration. 

Kind regards, also on behalf of the co-authors 

Meredith L. Overman, MSc

Department of Health Promotion, NUTRIM School of Nutrition and Translational Research in Metabolism, Maastricht University, Maastricht, Netherlands.

m.overman@maastrichtuniversity.nl

---

## [Decision Letter · Decision Letter 1]

28 Mar 2024

The cross-sectional association of parental psychosocial status with children’s Body Mass Index z-score and the mediating role of children’s energy balance behaviors - the ABCD Study

PONE-D-23-29554R1

Dear Dr. Overman,

We’re pleased to inform you that your manuscript has been judged scientifically suitable for publication and will be formally accepted for publication once it meets all outstanding technical requirements.

Kind regards,

Henri Tilga, PhD

Academic Editor

PLOS ONE

Additional Editor Comments (optional):

Reviewers' comments:

Reviewer's Responses to Questions

**Comments to the Author**

1. If the authors have adequately addressed your comments raised in a previous round of review and you feel that this manuscript is now acceptable for publication, you may indicate that here to bypass the “Comments to the Author” section, enter your conflict of interest statement in the “Confidential to Editor” section, and submit your "Accept" recommendation.

Reviewer #1: All comments have been addressed

2. Is the manuscript technically sound, and do the data support the conclusions?

Reviewer #1: Yes

3. Has the statistical analysis been performed appropriately and rigorously? 

Reviewer #1: Yes

4. Have the authors made all data underlying the findings in their manuscript fully available?

Reviewer #1: Yes

5. Is the manuscript presented in an intelligible fashion and written in standard English?

Reviewer #1: Yes

6. Review Comments to the Author

Reviewer #1: Dear PLOS ONE editor and authors

My apologies, it appears I have made several misreadings.

The authors have improved their manuscript, so I have no further comments.

7. PLOS authors have the option to publish the peer review history of their article (what does this mean?). If published, this will include your full peer review and any attached files.

Reviewer #1: No
